# Leptospirosis in pregnancy: A systematic review

**Sujitha Selvarajah**[1]*, **Shaolu Ran**[2], **Nia Wyn Roberts**[3], **Manisha Nair**[1]

**1** National Perinatal Epidemiology Unit, Nuffield Department of Population Health, University of Oxford, United Kingdom, **2** King's College Hospital NHS Foundation Trust, London, United Kingdom, **3** Bodleian Health Care Libraries, University of Oxford, Oxford, United Kingdom

* sujitha.selvarajah@nhs.net

## Abstract

### Introduction

Leptospirosis is a leading zoonotic disease worldwide with more than 1 million cases in the general population per year. With leptospirosis being an emerging infectious disease and as the world's environment changes with more floods and environmental disasters, the burden of leptospirosis is expected to increase. The objectives of the systematic review were to explore how leptospirosis affects pregnancy, its burden in this population, its effects on maternal and fetal outcomes and the evidence base surrounding treatment options.

### Methods

We performed a systematic review of published and unpublished literature using automated and manual methods to screen nine electronic databases since inception, with no language restriction. Two reviewers independently screened articles, completed the data extraction and assessment of risk of bias. Due to significant heterogeneity and paucity of data, we were unable to carry out a meta-analysis, but we conducted a pooled analysis of individual patient data from the case reports and case series to examine the patient and disease characteristics, diagnostic methods, differential diagnoses, antibiotic treatments, and outcomes of leptospirosis in pregnancy. The protocol for this review was registered on the International Prospective Register of Systematic Reviews, PROSPERO: CRD42020151501.

### Results

We identified 419 records, of which we included eight observational studies, 21 case reports, three case series and identified four relevant ongoing studies. Overall the studies were with moderate bias and of 'fair' quality. We estimated the incidence of leptospirosis in pregnancy to be 1.3 per 10,000 in women presenting with fever or with jaundice, but this is likely to be higher in endemic areas. Adverse fetal outcomes were found to be more common in pregnant patients who presented in the second trimester compared with patients who presented in the third trimester. There is overlap between how leptospirosis presents in pregnancy and in the general population. There is also overlap between the signs, symptoms and biochemical disturbances associated with leptospirosis in pregnancy and the

**Data Availability Statement:** All relevant data are within the manuscript and its Supporting Information files.

**Funding:** The systematic review and MN was funded by a Medical Research Council Career Development Award to Manisha Nair (Grant Ref:

MR/P022030/1). The funders had no role in the study design, data collection, analysis or writing of the report.

**Competing interests:** The authors have declared that no competing interests exist.

presentation of pregnancy associated conditions, such as Pre-Eclampsia (PET), Acute Fatty Liver of Pregnancy (AFLP) and HELLP Syndrome (Haemolysis Elevated Liver enzymes Low Platelets). In 94% of identified cases with available data, there was an indicator in the patient history regarding exposure that could have helped include leptospirosis in the clinician's differential diagnosis. We also identified a range of suitable antibiotic therapies for treating leptospirosis in pregnancy, most commonly used were penicillins.

## Conclusion

This is the first systematic review of leptospirosis in pregnancy and it clearly shows the need to improve early diagnosis and treatment by asking early, treating early, and reporting well. Ask early—broaden differential diagnoses and ask early for potential leptospirosis exposures and risk factors. Treat early—increase index of suspicion in pregnant patients with fever in endemic areas and combine with rapid field diagnosis and early treatment. Report well—need for more good quality epidemiological studies on leptospirosis in pregnancy and better quality reporting of cases in literature.

## Author summary

There are more than 1 million cases of leptospirosis in the general population each year. Leptospirosis is an emerging infectious disease and as the world's environment changes with more floods and environmental disasters, the impact of leptospirosis on the world is expected to increase. Leptospirosis is a zoonotic disease passed onto humans and can cause a range of illness from mild symptoms to severe organ failure and death. It is typically underreported and understudied, hence its classification as a 'Neglected Tropical Disease'. This is the first systematic review on Leptospirosis in Pregnancy looking at how common it is in pregnancy, how it affects maternal and fetal outcomes, and options for management. We found there to be overlap between how leptospirosis presents in the general population and in pregnancy, and that it can mimic non-infectious conditions that only present in pregnancy such as Pre-Eclampsia, Acute Fatty Liver of Pregnancy and other syndromes where liver and platelet function is affected. Adverse fetal outcomes were found to be more common in pregnant patients who presented in the second trimester compared with patients who presented in the third trimester. In 94% of identified cases with available data, there was a clue in the patient's history that could have indicated possible exposure to leptospirosis, which is very important in raising suspicion of a diagnosis of leptospirosis in pregnancy. We also identified a range of suitable antibiotic therapies for treating leptospirosis in pregnancy, most commonly used were penicillins. Our recommendations are: Ask early—broaden differential diagnoses and ask early for potential leptospirosis exposures and risk factors. Treat early—combine considering leptospirosis as a cause of fever in pregnant patients in endemic areas with prompt diagnosis and treatment. Report well—there is a need for more good quality epidemiological studies on leptospirosis in pregnancy and better quality reporting of cases in literature.

## Introduction

Over time, public health interest and the global burden of disease has shifted more towards non-communicable diseases, and away from communicable diseases [1]. Within this context, the COVID-19 pandemic has cast an abrupt and stark spotlight on the power of communicable and zoonotic disease over humanity [2]. Leptospirosis is both a 'neglected tropical disease' (NTD) and one of the most prevalent zoonotic diseases worldwide, with more than 1 million cases in the general population per year [3]. With global shifts in demography and climate, for example with two thirds of the global population expected to be living in urban areas by 2050 [4], as well as increasing environmental disasters and rising temperatures [5], the incidence of leptospirosis is forecast to increase [5–7]. With the commonest risk factors being exposure to outdoor work, floodwaters, and recreational water activities, we are already seeing a re-emergence of leptospirosis outbreaks following disasters such as floods [8]. As all mammals can be hosts to leptospirosis, it has a worldwide distribution, with outbreaks mainly in tropical and subtropical regions [8]. Leptospirosis in humans can cause a spectrum of clinical presentations; the majority of cases are subclinical [7] but it can also cause life threatening conditions like pulmonary haemorrhage, meningitis and respiratory distress syndrome [6,7].The severe form of the disease has been given the eponymous name 'Weil's disease' and is classically associated with life threatening renal and liver failure [9]. Given that leptospirosis is more common in men [6], due to a range of factors such as occupational exposure [10], and the combination of its NTD status, it is perhaps unsurprising that leptospirosis in pregnancy has not yet been extensively studied. This is the first systematic review exploring leptospirosis in pregnancy. As a treatable infectious disease, a better understanding of its incidence, presentation, and effects in pregnancy, as well as management options can significantly reduce mortality and morbidity from leptospirosis in pregnancy.

Leptospirosis cases generally and in pregnancy are thought to be underreported [6,11]. This can be partly attributed to its subtle presentations but also how its clinical presentation can mimic pregnancy associated conditions such as HELLP Syndrome (Haemolysis, Elevated Liver enzymes and Low Platelets, PET (Pre-Eclampsia) and AFLP (Acute Fatty Liver of Pregnancy), therefore further contributing to the issue of underdiagnosis [11]. In undertaking this systematic review of leptospirosis in pregnancy, this paper aims to answer three questions:

1. What is the incidence of leptospirosis in pregnant people, globally?

2. What are the maternal and fetal/infant outcomes of leptospirosis during pregnancy?

3. What is the evidence related to the efficacy and safety of drugs for treating leptospirosis in pregnancy?

## Methods

### Protocol and registration

The study is registered with PROSPERO: CRD42020151501 and the protocol can be found online https://www.crd.york.ac.uk/prospero/display_record.php?RecordID=151501. (S1 Text) The systematic review was carried out and presented in accordance with the Preferred Reporting Items for Systematic Reviews and Meta-Analyses (PRISMA) guidelines (Fig 1).

### Search and study selection

The final search was undertaken in October 2020 and updated in April 2021. We performed a systematic review of published and unpublished literature using automated and manual methods to screen the following nine electronic databases since inception, with no language restriction:

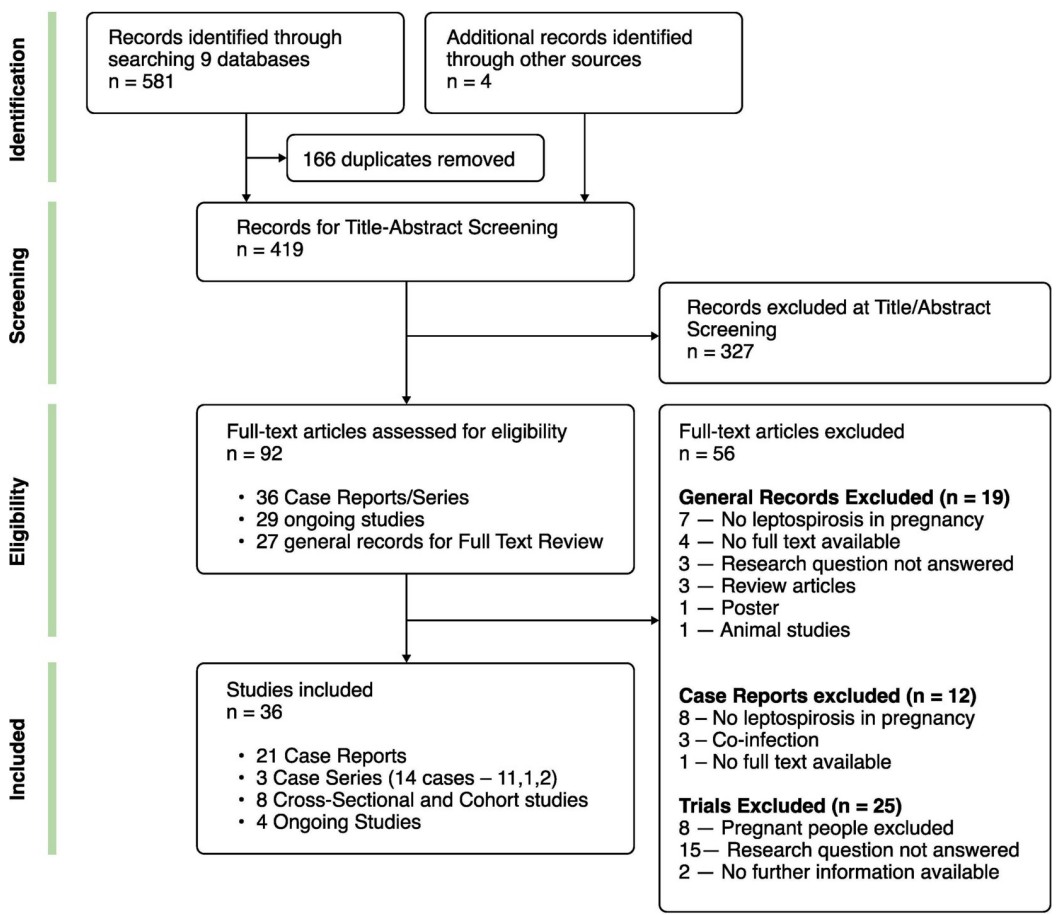

**Fig 1. Preferred Reporting Items for Systematic Reviews and Meta-Analyses (PRISMA) flow diagram of the study selection process.**

MEDLINE, Embase, CINAHL (Cumulative Index to Nursing and Allied Health Literature), Global Health (OvidSP), Web of Science and Cochrane Central Register of Controlled Trials (CENTRAL), 'Proquest Dissertations & Theses' and the trial registers www.ClinicalTrials.gov and www.who.int/trialsearch/. We also manually searched reference lists of included studies and key journals in the fields of obstetrics and infectious diseases. Our search strategy is outlined in S2 Text.

The searches were carried out twice independently by SS and NW. Bibliographic database Endnote, was used to manage references, identify duplicates and share references among the reviewers. The screening was carried out independently by the two reviewers SS and SR in two stages–(i) screening of titles and abstracts based on the pre-specified inclusion and exclusion criteria; (ii) screening of full-texts of the papers included during stage-1. The initial concordance rate at title/abstract review between the reviewers was 95.01%, and 91.49% at full text review with the average concordance rate being 93.25%. Any disagreements on inclusion of studies were discussed and, where possible, resolved by consensus after referring to the review protocol; a third reviewer MN was consulted for input on certain decisions. A record of decisions made for each article was maintained and a list of all excluded articles after full text review is provided in S5 Table. The two reviewers also independently extracted data from all included papers, before reviewing data congruence. Discordance was again discussed and resolved via the review protocol or third reviewer MN consultation.

The inclusion criterion was cases of leptospirosis in pregnancy and exclusion criteria being non-pregnant population diagnosed with leptospirosis or unconfirmed diagnosis of leptospirosis. We included all study designs except ecological studies, expert opinion and those describing mainly the pathology and pathogenesis of the infections. We also excluded papers focusing on preventative measures eg. vaccinations or travel advice for pregnant women, as these did not answer the research questions. We initially excluded case reports and series and categorised them as such, but later decided to systematically include, extract data from, synthesise and present findings. This was due to a substantial proportion of the relevant evidence being identified as case reports. Given the sparse literature on this topic generally and exemplified by our review, we felt the case reports and series could provide vital information in answering our research questions. A record of all included case reports and series can be found in S4 Table with the details of the data extraction process available in S3 Text. Furthermore, we initially excluded correspondence, however after identifying a relevant case report that had been submitted as correspondence [12], we reran the searches and searched all correspondence for relevant literature —no further relevant correspondence records were identified.

## Quality assessment

The risk of bias across studies was assessed using the National Heart, Lung and Blood Institute (NHLBI) Quality Assessment Framework to assess Cohort and Cross-Sectional studies (S2 Table). A modified version of the NHLBI Framework for Case Series [13], incorporating components from an existing alternative criteria for appraising case reports was used [14], thus making it suitable to assess bias in both case reports and series. The modified quality assessment tool can be found in S1 Table. Both SS and SR independently reviewed the quality of each included paper using these quality assessment tools, and agreed on each criteria with consensus.

## Analysis

We undertook a formal narrative synthesis to answer each research question. We analysed the case reports, which do not have a comparator group, separately. We estimated the pooled incidence rate with 95% Exact Binomial Confidence Interval (CI) of leptospirosis in pregnancy if there were two or more studies with comparable denominators that could be included in the analysis. We planned to use a random-effects model/Peto method to calculate pooled risk ratios/ odds ratios (with 95% CI) to compare maternal and child outcomes in women diagnosed with leptospirosis during pregnancy compared with healthy controls, and effectiveness of drugs (antibiotics, mainly). Due to a paucity of studies and high heterogeneity of included studies, this was not possible. Instead we conducted a pooled analysis of individual patient data (IPD) by collating the data from the case reports and case series to examine the patient and disease characteristics, diagnostic methods, differential diagnosis, antibiotic treatments, and outcomes of leptospirosis in pregnancy. The methodology was similar to another systematic review of case reports [15]. All analyses were conducted using Stata SE, v15 (StataCorp, College Station, Texas).

We planned to use a standard funnel plot to analyse publication bias if 5 or more studies were included, as per Cochrane recommendation, but this was not possible for the case reports and case series.

## Results

### Study characteristics

Fig 1 summarises our identification, screening and selection process. We reviewed a total of 419 records across nine databases (Fig 1). After screening and full-text review, we included

eight observational studies—six of which were cross-sectional and two were cohort studies and 35 cases of leptospirosis in pregnancy from 21 case reports and three case series (11 cases from case-series-1, one case from case-series-2, two cases from case-series-3). Twenty-four of the included articles were in English, three in French, two in German, as well as one Spanish, one Italian, and one Russian. The non-English articles were translated prior to data extraction. We did not identify any unpublished literature.

We identified four relevant ongoing studies summarised in S3 Table. The ongoing studies could potentially answer our research questions, but none of these have published data at present. Out of 25 initially identified ongoing studies, 15 were excluded as they did not answer the research questions and two were excluded as further information was not available (S5 Table). Of note is that eight studies were not included in the review because those who were pregnant were excluded from these studies. This highlights the importance of including pregnant individuals in studies where possible, so we can have an appropriate evidence-base to draw upon in clinical practice.

**Date of publication.** Of the 32 published articles, 17 were published after 2000 (53%), seven between 1980 and 2000 (22%), five between 1960 and 1980 (16%), and three published before 1960 (9%). Six of the eight observational studies were published after 2000. One of the case series, with 11 cases, was from French Guyana, published in 1995 but gathered data from 1979–1981 [16].

**Geographical distribution and population characteristics.** For purposes of summarising the geographical distribution of the studies, we categorised them using the World Health Organization regions. Of the 32 articles, 14 were published in Europe (44%), six in the Western Pacific (19%), five in the Americas region (16%), four in the South-East Region (13%), two in the Eastern Mediterranean (6%), and only one from the African region (3%). Three of the ongoing studies are based in Europe, with one in the Western Pacific Region.

The geographical distribution and type of the studies identified are summarised in Fig 2 and the population characteristics of the included cohort and cross-sectional studies are summarised in Table 1. For the case reports, the majority of cases were in urban areas (70%, n = 23). One cross sectional study (Study 4, Table 1) based in Yucatan, Mexico found a higher frequency of leptospirosis in pregnancy leading to spontaneous miscarriage in rural areas (4%) compared to urban (1.3%) [17].

**Quality of the included studies.** The majority of observational studies had a moderate to high level of bias, being of 'fair' to 'poor' quality, except for one cross-sectional study (Study 1, Tables 1 and 2) which was graded to be of 'good' quality. For the case reports and series, the majority of cases were graded as moderate to low bias, giving the overall quality of evidence for this part of the review as 'fair' to 'good' (Table 3). Given the range of quality across the different types of studies, the articles in this review are overall of moderate bias and of 'fair' quality. The details of the quality assessment are summarised in Tables 2 and 3, with the full criteria available in S1 and S2 Tables.

**The global burden of leptospirosis in pregnancy.** Calculating a global incidence rate for leptospirosis in pregnancy was not possible. This was due to a general paucity of data on leptospirosis in pregnancy, combined with the heterogeneity of the study population which provided the denominator for the rates. Two studies looked at seroprevalence of leptospirosis in febrile pregnant women [18,20], two studies at seroprevalence of leptospirosis in jaundiced pregnant women [19,22], one explored cases of leptospirosis in women who had spontaneous abortion [17], another on leptospirosis as a cause of abortion, stillbirth or developmental anomalies [24]. One cross-sectional study looked at seroprevalence of leptospirosis in 442 healthy pregnant women across 10 Caribbean countries [17,20,21], and one investigating causes of fever and pulmonary syndrome in pregnant women, where no cases of leptospirosis were identified [22,23]. Pooling together two datasets (Studies 1,3; Table 1 [18,20]), we

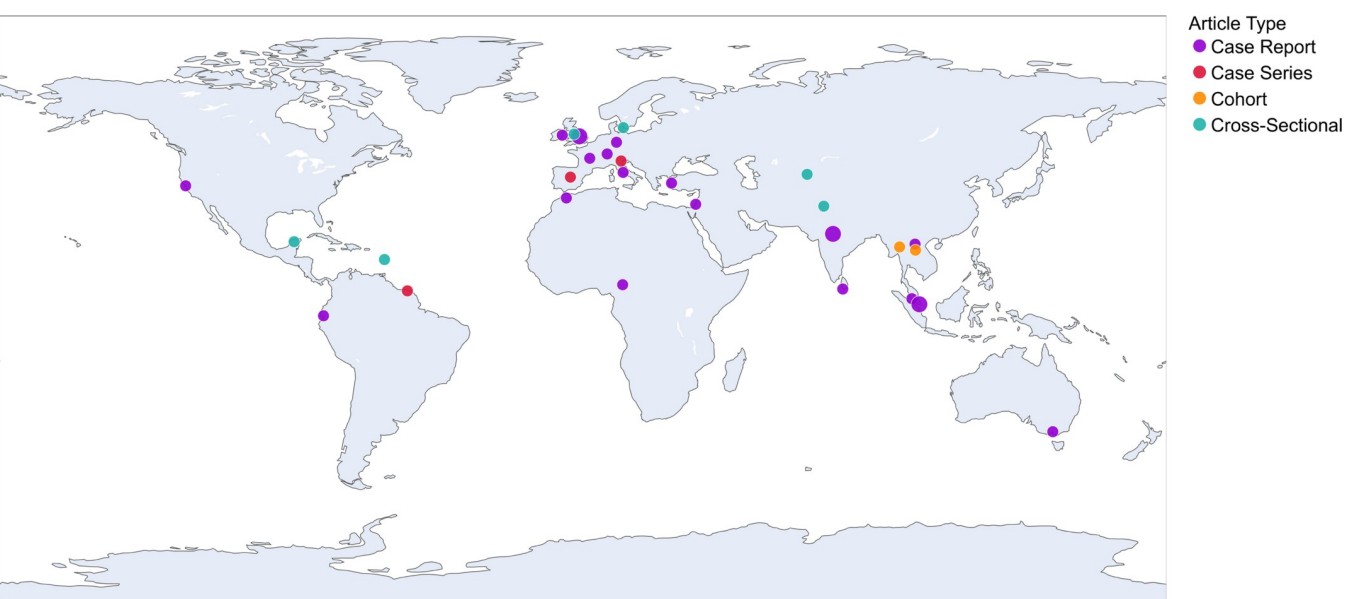

**Fig 2. Geographical distribution of included published articles.** The colour corresponds to the type of article (see key), and the size of the marker corresponds to the number of articles published in that country. There were no more than two articles of the same type from one country. Bubble map created with Plotly Graphing Libraries - Made with Natural Earth. Free vector and raster map data @ naturalearthdata.com.

calculated the incidence of leptospirosis in febrile pregnant patients to be 1.3 per 10,000 (95% CI 0.5 to 2.9 per 10,000). The calculated incidence of leptospirosis in pregnant patients with jaundice, pooling two datasets (Studies 2, 6; Table 1 –[21,24]; Table 1) was comparable, 1.3 per 10,000 (95% CI 0.2 to 4.6 per 10,000).

The study looking at the seroprevalence of zoonotic pathogens in 442 healthy pregnant women across 10 Caribbean countries by measuring immunoglobulin G (IgG) antibodies estimated a seroprevalence of 18.6% +/- 3.6 overall, but importantly all 10 countries had seropositive pregnant women suggesting endemicity of leptospirosis in the region [21].

**Maternal and infant outcomes of leptospirosis during pregnancy.**   Of the eight cross-sectional and cohort studies, three of the studies [22–24] did not identify any cases of leptospirosis in pregnancy, so were not helpful in answering the research questions regarding outcomes or treatment. Of the remaining five studies, only three commented on outcomes associated with leptospirosis in pregnancy, which included spontaneous miscarriage [17], maternal death [19], term birth with the baby being small for gestational age [20]. One study did not observe any adverse maternal or fetal outcomes in the five women with leptospirosis who gave birth to term, normal weight infants [18,20].

While there was limited information in the observational studies, we were able to use data from the case reports and case series to answer the second research question on maternal and fetal/infant outcomes associated with leptospirosis in pregnancy. The epidemiological, patient and pregnancy characteristics of the cases are summarised in Table 4.

The median maternal age was 29.5, with the interquartile range (IQR) being 9.75. Where the data was available, most pregnant patients were healthy, with only three patients having medical comorbidities, and these were obesity and dental caries for one [38], perioral herpes for another [43], and diet-controlled gestational diabetes mellitus for the other [44]. The median gestation of presentation with symptom(s) was 27 weeks, with an IQR of 10.5. No patients presented in the first trimester, 43% presented in the second trimester (n = 15) and 57% presented in the third trimester (n = 20).

**Table 1. Summary of the observational studies.**

| Study (Language) | Study Design | Location/ WHO Region | Aim | Population/ Time period | Maternal Outcomes | Neonatal Outcomes | Number of Leptospirosis Cases/ Seroprevalence | Denominator | Outcomes associated with Leptospirosis | Treatment | Quality |
|---|---|---|---|---|---|---|---|---|---|---|---|
| Study 1 [18] Rose McGready et al. (English) | Prospective Cohort | North Western Thai-Burmese Border SEAR | To investigate aetiology of fever in pregnant women and associated pregnancy outcomes | 409 pregnant women, any gestation attending antenatal clinic and febrile (>37.5 °C) May 2004 – January 2006 | 1.Fever Diagnosis 2. Maternal mortality 3. Concomitant Infection | 1. Miscarriage (birth <28/40 weeks) 2. Stillbirth (birth >28/40 weeks with no signs of life) 3. Congenital abnormality 4. Birth Weight 5. Gestational Age 6. Neonatal mortality | 5/203 | 203 febrile pregnant women | No adverse maternal or infant outcomes reported Mean Estimated Gestational Age at Birth: 40.2 ± 0.9 weeks Mean Birth Weight: 3148 ± 580 gms | Azithromycin (ambulant + febrile) Ceftriaxone + Metronidazole + Azithromycin (severely ill + febrile) | Good– 12 |
| Study 2 [19] Ayesha Shafaq et al. (English) | Cross-Sectional | Lahore, Pakistan– Mayo Hospital EMR | Find fetomaternal outcomes in jaundice complicating pregnancy | 49 jaundiced pregnant women attending clinic regularly March–November 2018 | 1. Preeclampsia 2. Eclampsia 3. Acute Renal Failure 4. Disseminated Intravascular Coagulopathy 5. Sickle Cell Crisis 6. Postpartum haemorrhage 7. Multi-organ Failure 8. Fever 9. Shock 10. Death | Nil | 2/50 | 50 Jaundiced pregnant patient | Death 1/2 of leptospirosis cases | NS | Poor– 3 |
| Study 3 [20] Vilada Chansamouth et al. (English) | Prospective Cohort | Vientiane, Laos–Mahosot Hospital, Mother and Child Hospital WPR | To determine causes and impact of fever in pregnant women admitted to 2 central hospitals in Laos | 250 pregnant women with temp ≥37.5 °C February 2006 –November 2010 | 1. Fever Diagnosis 2. Maternal mortality | 1. Preterm birth (<37/40 weeks) 2. term but low birth weight (<2500gms) 3. miscarriage (delivery <28/ 40 weeks gestation) 4. Stillbirth (delivery > 28/ 40 weeks with no signs of life) 5. neonatal mortality | 1/250 | 250 febrile pregnant women | 1. Gestational Age 38/40 weeks 2. Term but low birth weight (2000gms) No other adverse outcomes | NS | Fair– 9 |

(*Continued*)

**Table 1.** (Continued)

| | Study (Language) | Study Design | Location/ WHO Region | Aim | Population/ Time period | Maternal Outcomes | Neonatal Outcomes | Number of Leptospirosis Cases/ Seroprevalence | Denominator | Outcomes associated with Leptospirosis | Treatment | Quality |
|---|---|---|---|---|---|---|---|---|---|---|---|---|
| Study 4 [17] | Cárdenas-Marrufo MF et al. (English) | Cross-Sectional | Yucatan, Mexico. Hospital Communitaro in City of Ticul and Hospital Materno Infantil in City of Merida AMR | To estimate the frequency of Leptospira interrogans infection in women with spontaneous abortion in the state of Yucatan, Mexico | 81 women who had spontaneous abortions February–June 2009 | 1. Serological confirmation of Leptospirosis | 1. Spontaneous Miscarriage/ Abortion | 11/81 (13.6%) diagnosed on MAT. 2 of these 11 also positive on ELISA IgM | 81 Women who had spontaneous abortions | Higher frequency of serologically confirmed leptospirosis in rural vs urban hospital (12.3% vs 1.3%) | NS | Fair– 6 |
| Study 5 [21] | Heidi Wood et al. (English) | Cross-Sectional | 10 Caribbean countries: Antigua-Barbuda, Belize, Bermuda, Dominica, Grenada, Jamaica, Montserrat, St. Kitts-Nevis, St. Lucia, and St. Vincent-Grenadines AMR | Estimate of seroprevalence of 7 zoonotic pathogens in pregnant population within 10 Caribbean countries | 442 healthy pregnant women from the 10 countries and up to 50 women from each country 2009–2011 | 1. Seroprevalence of 7 zoonotic pathogens in pregnant women from 10 Caribbean Islands; Leptospirosis, Hepatitis E virus, dengue virus, hantaviruses, spotted fever group rickettsiae (SFGR), typhus group rickettsiae (TGR), and Coxiella burnetii | Nil | Zoonotic pathogen seroprevalence (%) +/- 95% CI: all countries 18.6 ± 3.6 Antigua-Barbuda 10.5 ± 9.8 Belize 8.0 ± 7.5 Bermuda 12.0 ± 9.0 Dominica 16.0 ± 10.2 Dominica 21.6 ± 11.3 Jamaica 42.6 ± 14.1 Montserrat 20.0 ± 20.2 St Kitts-Nevis 9.1 ± 8.5 St Lucia 19.6 ± 11.5 St Vincent-Grenadines 26.0 ± 12.2 | 442 healthy pregnant women | Frequency of seroprevalence of each Caribbean country | NS | Poor– 2 |

(Continued)

**Table 1.** (Continued)

| | Study (Language) | Study Design | Location/ WHO Region | Aim | Population/ Time period | Maternal Outcomes | Neonatal Outcomes | Number of Leptospirosis Cases/ Seroprevalence | Denominator | Outcomes associated with Leptospirosis | Treatment | Quality |
|---|---|---|---|---|---|---|---|---|---|---|---|---|
| Study 6 [22] | P. Friedlander et al. (English) | Cross-Sectional | Copenhagen, Denmark EUR | To identify causes of icterus in pregnancy and how this affects treatment and prognosis for mother and child. | 135 pregnant women with icterus Copenhagen, Denmark 1950–1962 | 1. Liver Functions Tests 2. Anaemia 3. Toxaemia 4. Pyuria 5. History of itching 6. History of hepatitis contact 7. Caesarean section 8. Instrumental delivery 9. Induced labour 10. Post-Partum Haemorrhage | 1. Prematurity 2. Perinatal mortality 3. Birth Weight (<2500gms) 3. Congenital abnormalities | 0 | 103 jaundiced pregnant women | Nil | NS | Poor–2 |
| Study 7 [23] | An, A. V. et al) (Russian) | Cross-Sectional | Tashkent, Uzbekistan EUR | Find out virologic causes of infectious diseases in pregnant women with fever and pulmonary syndrome, given seasonality and treatment ineffectiveness of antibacterial/ antifungal medications | 31 blood serum samples from sick pregnant patients who had fever and pulmonary syndrome. 2007 | Nil | 1. Confirmed virological infection on PCR | 0 | 31 febrile pregnant women with pulmonary syndrome | Nil | NS | Poor–3 |

*(Continued)*

**Table 1.** (Continued)

| | Study (Language) | Study Design | Location/ WHO Region | Aim | Population/ Time period | Maternal Outcomes | Neonatal Outcomes | Number of Leptospirosis Cases/ Seroprevalence | Denominator | Outcomes associated with Leptospirosis | Treatment | Quality |
|---|---|---|---|---|---|---|---|---|---|---|---|---|
| Study 8 [24] | Sanghi A et al. (English) | Cross-Sectional | Sharoe Green Hospital, Preston, and Chorley District General Hospital, Chorley, Lancashire EUR | To investigate infectious disease causes of mid trimester miscarriage, stillbirth, or termination for developmental or chromosomal abnormalities. | 136 women from an urban, rural and farming community who had mid-trimester miscarriage, stillbirth, or termination for developmental or chromosomal abnormalities, but no obvious signs of infection. 1992–1993 | 1. Incomplete, inevitable or missed abortion at 2–12 weeks gestation 2. Stillbirth from 24–42 weeks 3. Termination after 12 weeks of pregnancy for a developmental or chromosomal anomaly | 1. Trisomy 2. Large Ventricular Septal Defect 3. Myotonic Dystrophy 4. Gastroschisis 5. Vater Syndrome 6. Neuromuscular 7. Neural Tube Defect 8. Multiple Anomalies 9. Fetal Limb Anomaly 10. Down's Syndrome 11. Turner's Syndrome | 0 | 136 women presenting either incomplete, inevitable or missed abortion at > = 12 weeks gestation or a still birth 24–42 weeks, or had a termination after 12 weeks of pregnancy for a developmental or chromosomal anomaly | Nil | NS | Fair– 6 |

NS = Not Specified

**Table 2. Quality Assessment of Observational Studies.**

| | | Study 1 [18] | Study 2 [19] | Study 3 [20] | Study 4 [17] | Study 5 [21] | Study 6 [22] | Study 7 [23] | Study 8 [24] |
|---|---|---|---|---|---|---|---|---|---|
| | | R. McGready et al. | A. Shafaqet et al. | V. Chansamouth et al. | Cárdenas-Marrufo MF et al. | H. Wood et al. | P. Friedlander et al. | An, A. V. et al | Sanghi A et al. |
| **Quality Assessment Criteria** | 1 | Y | Y | Y | Y | Y | Y | Y | Y |
| | 2 | Y | Y | Y | Y | N | N | Y | Y |
| | 3 | Y | NA | NA | Y | N | N | NA | NA |
| | 4 | Y | Y | Y | N | N | N | NA | Y |
| | 5 | Y | N | NA | N | N | N | N | N |
| | 6 | Y | N | Y | N | N | N | N | N |
| | 7 | Y | N | Y | N | N | N | NA | N |
| | 8 | NA | N | NA | Y | N | N | N | Y |
| | 9 | Y | N | Y | Y | N | N | NA | Y |
| | 10 | Y | N | Y | NA | N | N | N | NA |
| | 11 | Y | N | Y | Y | Y | Y | Y | Y |
| | 12 | N | N | N | Y | NA | NA | N | N |
| | 13 | Y | NA | Y | NA | NA | NA | NA | NA |
| | 14 | Y | N | N | N | NA | NA | N | NA |
| **Quality Rating** | | 12 | 3 | 9 | 7 | 2 | 2 | 3 | 6 |
| **Quality Category** | | G | P | F | F | P | P | P | F |

**Summary: Quality Assessment of Observational Studies**

| | Quality Assessment Rating | Quality Assessment Category | n = |
|---|---|---|---|
| | <5 | Poor Quality (P) | 4 |
| | 5–9 | Fair Quality (F) | 3 |
| | 10–14 | Good Quality (G) | 1 |

The clinical characteristics of leptospirosis in pregnancy are summarised in Table 5, including signs and symptoms, bedside and biochemical investigations, as well as reporting on cardiovascular, respiratory and liver function disturbances. The most common presenting symptom was fever with 91% of patients presenting febrile (n = 31). Other symptoms patients had were myalgia (59%, n = 17), jaundice (47%, n = 16), malaise (77%, n = 10), headache (50%, n = 11) abdominal pain (43%, n = 10) and nausea and/or vomiting (45%, n = 10). None of the women for whom relevant information was available, presented with visual disturbances. When comparing differentials for fever, conjunctival suffusion, a sign that can sometimes be used to differentiate between leptospirosis and other causes of febrile illness such as malaria [48], could be helpful in the context of pregnancy in helping distinguish between pregnancy related disorders such as PET and HELLP and leptospirosis. Information on ocular examination was only available for 13 out of the 35 cases, and of these, 38% (n = 5) of the women had conjunctival suffusion. Complete data on all variables was not available from the included case reports and percentages were calculated after excluding missing data.

From the available data, we found that nine patients had a leucocytosis (60%), 10 had a thrombocytopaenia (71%), 15 (63%) whose kidney function was affected with raised creatinine and/or oliguria and the majority had abnormal LFTs of some form (89%, n = 24). Of the 24 with the abnormal LFTs, 19 had a transaminitis, 20 had hyperbilirubinaemia, 10 had a coagulopathy. We were unable to categorise and grade severity of the transaminitis due to detailed information not being available for 10 of the 19 transaminitis cases.

**Table 3. Quality Assessment of Case Reports and Case Series.**

| Quality Assessment Criteria | CR1 [25] | CR2 [26] | CR3 [27] | CR4 [28] | CR5 [29] | CR6 [30] | CR7 [31] | CR8 [32] | CR9 [33] | CR10 [34] | CR11 [35] | CR12 [36] | CR13 [37] | CR14 [38] | CR15 [39] | CR16 [12] | CR17 [40] | C18 [41] | CR19 [42] | CR20 [43] | CR21 [44] | CS1* [45] | CS2 [46] | CS3 [47] |
|---|---|---|---|---|---|---|---|---|---|---|---|---|---|---|---|---|---|---|---|---|---|---|---|---|
| 1 | Y | Y | Y | Y | Y | Y | Y | Y | Y | Y | Y | Y | Y | Y | Y | Y | Y | Y | Y | Y | Y | Y | Y | Y |
| 2 | Y | Y | Y | Y | Y | Y | Y | Y | N | N | Y | Y | Y | Y | Y | Y | Y | Y | N | Y | Y | Y | Y | Y |
| 3 | Y | N | Y | N | Y | Y | NA | Y | NA | NA | Y | N | NA | NA | Y | Y | Y | Y | N | Y | Y | N | Y | N |
| 4 | Y | N | Y | Y | Y | Y | Y | Y | N | N | Y | Y | N | Y | NA | Y | Y | Y | Y | Y | Y | Y | N | Y |
| 5 | Y | Y | Y | N | N | Y | N | Y | N | N | Y | N | Y | Y | N | Y | N | Y | N | Y | Y | Y | N | Y |
| 6 | Y | N | Y | N | N | N | N | Y | N | N | Y | N | N | N | N | Y | N | Y | N | N | Y | NA | N | N |
| 7 | NA | NA | NA | NA | NA | NA | NA | NA | NA | NA | NA | NA | NA | NA | NA | NA | NA | NA | NA | NA | NA | NA | NA | NA |
| 8 | NA | NA | NA | NA | NA | NA | NA | NA | NA | NA | NA | NA | NA | NA | NA | NA | NA | NA | NA | NA | NA | NA | Y | N |
| 9 | NA | NA | NA | NA | NA | NA | NA | NA | NA | NA | NA | NA | NA | NA | NA | NA | NA | NA | NA | NA | NA | NA | NA | NA |
| 10 | NA | NA | NA | NA | NA | NA | NA | NA | NA | NA | NA | NA | NA | NA | NA | NA | NA | NA | NA | NA | NA | NA | Y | Y |
| **Quality Rating** | 6 | 3 | 6 | 3 | 4 | 5 | 3 | 6 | 1 | 1 | 6 | 3 | 3 | 4 | 3 | 6 | 3 | 6 | 2 | 5 | 6 | 3 | 5 | 4 |
| **Quality Category** | G | F | G | F | F | G | F | G | P | P | G | F | F | F | F | G | F | G | P | G | G | P | F | F |

**Summary: Quality Assessment of Case Reports and Case Series**

| Case Reports | | | Case Series | | |
|---|---|---|---|---|---|
| Quality Assessment Rating | Quality Assessment Category | n = | Quality Assessment Rating | Quality Assessment Rating | n = |
| 0–2 | Poor Quality (P) | 4 | 0–3 | Poor Quality (P) | 0 |
| 3–4 | Fair Quality (F) | 9 | 4–6 | Fair Quality (F) | 2 |
| 5–6 | Good Quality (G) | 9 | 7–10 | Good Quality (G) | 0 |

The quality assessment criteria for the case reports and case series can be found in S1 Table and if it was a Case Report, only Criteria 1–6 were applicable. Criteria 7–10 were only applicable to Case Series (CS).

*CS1 –One of the two cases were excluded as it did not include leptospirosis in pregnancy and therefore CS1 was quality assessed using the case report criteria.

**Table 4. Characteristics of patients with leptospirosis in pregnancy.**

| Characteristics | | Frequency (n = 35)* | %* |
|---|---|---|---|
| **Type of residence** | Rural | 10 | 30% |
| | Urban | 23 | 70% |
| | NA | 2 | |
| **Geographical Region** | African Region (AFR) | 1 | 3% |
| | Region of the Americas (AMR) | 13 | 37% |
| | South-East Asia Region (SEAR) | 3 | 9% |
| | European Region (EUR) | 12 | 34% |
| | Eastern Mediterranean Region (EMR) | 1 | 3% |
| | Western Pacific Region (WPR) | 5 | 14% |
| **Medical comorbidities** | Absent | 8 | 73% |
| | Present | 3 | 27% |
| | NA | 24 | |
| **Maternal age at presentation** | Median | 29.5 | |
| | IQR | 9.75 | |
| | NA | 2 | |
| **Gestation at presentation** | Median | 27 | |
| | IQR | 10.5 | |
| | NA | 1 | |

*Percentages were calculated after excluding missing data (NA); NA = Not Available

## Diagnosing Leptospirosis

Table 6 summarises the diagnostics, possible differential diagnoses and outcomes reported for cases of leptospirosis in pregnancy. All cases reported a serological confirmation of leptospirosis, 41% (n = 13) had the diagnosis made through an agglutination test, 53% (n = 17) with ELISA, 6% (n = 2) used both methods, and the method was not specified in two cases. For those cases where diagnosis was made with ELISA, we could derive that IgM antibodies were detected as this is how ELISA for leptospirosis diagnosis is usually carried out [49,50]. Taking this into account, out of the 35 cases, there were a total of 23 cases where IgM was detected, and 12 cases where it was not derivable if IgM was present and/or detected. These non-derivable results were either because the method of serological confirmation was not specified (n = 3) or it was confirmed with an agglutination test. There was only one case which mentioned the presence and confirmation of IgG antibodies [41]. With agglutination tests, both IgM and IgG antibodies are detected, but it is not possible to distinguish between them [50]. Furthermore for the agglutination tests, whilst titres were reported for some cases, we did not synthesise this data as agglutination titres cut-off values are dependent on local seroprevalence, with endemic areas having higher thresholds [50], and thus with our cases being reported across the world, from 1949 to now, it would be neither comparable nor useful. Leptospira IgM antibodies are produced around the first week [51,52], but can persist for months to years, so by itself cannot reliably be used to confirm acute infection [49,53]. It is recommended that a positive ELISA is confirmed by a MAT [50].

The data collected on blood culture confirmation in 10 cases showed that there were only two cases of confirmed leptospirosis on blood cultures and eight had sterile blood cultures. Blood culture diagnosis of leptospirosis infection is generally not the chosen method for rapid

**Table 5. Clinical characteristics of leptospirosis in pregnancy.**

| | | Frequency N = 35* | %* |
|---|---|---:|---:|
| **Signs and Symptoms** | | | |
| **Abdominal Pain** | Present | 10 | 43% |
| | Absent | 13 | 57% |
| | NA | 12 | |
| **Nausea/Vomiting** | Present | 10 | 45% |
| | Absent | 12 | 55% |
| | NA | 13 | |
| **Fever** | Present | 31 | 91% |
| | Absent | 3 | 9% |
| | NA | 1 | |
| **Clinical Jaundice** | Present | 16 | 47% |
| | Absent | 18 | 53% |
| | NA | 1 | |
| **Myalgia** | Present | 17 | 59% |
| | Absent | 12 | 41% |
| | NA | 6 | |
| **Malaise** | Present | 10 | 77% |
| | Absent | 3 | 23% |
| | NA | 22 | |
| **Peripheral Oedema** | Present | 3 | 50% |
| | Absent | 3 | 50% |
| | NA | 29 | |
| **Headache** | Present | 11 | 50% |
| | Absent | 11 | 50% |
| | NA | 13 | |
| **Visual Disturbances** | Present | 0 | 0% |
| | Absent | 3 | 100% |
| | NA | 32 | |
| **Conjunctival Suffusion** | Present | 5 | 38% |
| | Absent | 8 | 62% |
| | NA | 22 | |
| **Bedside Investigations** | | | |
| **Blood Pressure** | Hypotension | 2 | 17% |
| | Hypertension | 1 | 8% |
| | Normotensive | 9 | 75% |
| | NA | 23 | |
| **Tachycardia** | Present | 4 | 44% |
| | Absent | 5 | 56% |
| | NA | 26 | |
| **Oliguria** | Present | 6 | 30% |
| | Absent | 14 | 70% |
| | NA | 15 | |
| **Proteinuria** | Present | 11 | 48% |
| | Absent | 12 | 52% |
| | NA | 12 | |

(*Continued*)

**Table 5.** (Continued)

|  |  | Frequency N = 35[*] | %[*] |
|---|---|---|---|
| **Glasgow Coma Scale (GCS)** | Reduced GCS | 6 | 32% |
|  | GCS 15/15 | 13 | 68% |
|  | Not derivable | 16 |  |
| **Respiratory or Cardiovascular Abnormalities** |  |  |  |
| **Respiratory abnormalities** | Present | 9 | 64% |
|  | Absent | 15 | 36% |
|  | NA | 21 |  |
| **Cardiovascular abnormalities** | Present | 5 | 56% |
|  | Absent | 4 | 44% |
|  | NA | 26 |  |
| **Biochemical Investigations** |  |  |  |
| **White Blood Cells** | Leucocytosis | 9 | 60% |
|  | Normal Count | 6 | 40% |
|  | NA | 20 |  |
| **Platelets** | Thrombocytopaenia | 10 | 71% |
|  | Normal count | 4 | 29% |
|  | NA | 21 |  |
| **Creatinine** | Raised | 15 | 63% |
|  | Normal | 9 | 38% |
|  | NA | 11 |  |
| **Liver Function Tests** |  |  |  |
| **Liver Function Tests** | Abnormal | 24 | 89% |
|  | Normal | 3 | 11% |
|  | NA | 8 |  |
| **Transaminases** | Transaminitis | 3 | 14% |
|  | Normal | 19 | 86% |
|  | NA | 13 |  |
| **Bilirubin** | Hyperbilirubinaemia | 20 | 69% |
|  | Normal | 9 | 31% |
|  | NA | 6 |  |

[*]Percentages were calculated after excluding missing data (NA); NA = Not Available

diagnosis, for a number of reasons. Firstly, leptospira is a fastidious organism, requiring fast transport to a processing lab, and can take from 1–6 weeks for growth to be detectable and is therefore limited in making a quick diagnosis [54]. Secondly, the survival of leptospira in blood culture bottles can vary depending on the type of culture medium used [55]. Furthermore, leptospira in the blood tends to not be detectable after approximately 10 days, which is roughly when antibodies start to be detectable, so it might not be helpful in acute diagnosis [49].

There were six cases where leptospira was clearly identified in the urine, and just two cases where leptospira could be identified from placental tissue. Urinary confirmation of leptospirosis is again not a diagnostic method of choice due to limited survival of leptospira in human urine [50], with the low pH of human urine thought to be a key limiting factor [51]. Regarding testing for leptospira in placental tissue, we know that leptospira can cross the placenta and infect the fetus [28,51]. It could be that leptospira is unable to survive in autolysing tissue [31]

**Table 6. Summary of diagnostics, differential diagnoses and outcomes reported for cases of leptospirosis in pregnancy.**

| Methods used to diagnose leptospirosis in pregnant women | | Frequency n = 35* | %* |
|---|---|---|---|
| **Leptospirosis Exposure (from patient history)** | Present | 17 | 94% |
| | Absent | 1 | 6% |
| | NA | 17 | |
| **Method of serological confirmation** | Agglutination Test | 13 | 41% |
| | ELISA | 17 | 53% |
| | Both | 2 | 6% |
| | NA | 3 | |
| **Blood Culture** | Sterile | 8 | 80% |
| | Leptospira confirmed | 2 | 20% |
| | NA | 25 | |
| **Reported differential diagnoses of leptospirosis in pregnancy** | | | |
| **PET** | Yes | 3 | 25% |
| | No | 9 | 75% |
| | NA | 23 | |
| **AFLP** | Yes | 6 | 40% |
| | No | 9 | 60% |
| | NA | 20 | |
| **HELLP** | Yes | 9 | 64% |
| | No | 5 | 36% |
| | NA | 21 | |
| **Maternal and Fetal outcomes of leptospirosis in pregnancy** | | | |
| **Fetal Outcomes** | Fetal Death/IUFD/Miscarriage/Stillborn | 20 | 57% |
| | Neonatal death (after live birth) | 1 | |
| | Live birth | 15 | 43% |
| | Of those born live (n = 15) | | |
| | Preterm birth | 8 | 53% |
| | Term birth | 7 | 47% |
| **Maternal Outcomes** | Mother died | 4 | 11% |
| | Mother survived | 31 | 89% |

*Percentages were calculated after excluding missing data (NA); NA = Not Available; PET = Pre-Eclampsia; AFLP = Acute Fatty Liver of Pregnancy; HELLP = Haemolysis, Elevated Liver enzymes, Low Platelets.

that leads to leptospira often not being isolated in placental tissue. The French case series tested three cases for leptospira in placental tissue (out of 16), resulting in two positive cases [28,51]. The majority of other case reports did not report testing for placental leptospira.

Interestingly for 17 cases (94%), there was a potential exposure to leptospirosis identified from the history taken by the healthcare providers. Whilst we cannot tell if these questions were asked retrospectively after serological confirmation, such information about potential exposures can be useful for more nuanced differential diagnoses that include infectious diseases, and in particular, leptospirosis. Exposures identified from histories taken in the case reports included eating raw grass and vegetables, recent travel to an endemic area, visiting nearby public baths, exposure to farm animals and generally being in farming environments

from paddy fields to working in a milk parlour, exposure to rat infestations, either at home or at establishments where they had visited.

**Differential diagnoses.**   In investigating a rare condition in pregnancy which causes liver dysfunction, we also explored the overlap in how leptospirosis presents and affects outcomes, compared to the more common disorders of pregnancy which cause liver dysfunction, namely, pre-eclampsia (PET), HELLP syndrome and AFLP. It has been documented that leptospirosis can mimic these disorders in pregnancy [11].

There was a distinct paucity of data for this section, for example only 12 of the 35 cases mentioned blood pressure of the patient, meaning the dataset for ascertaining if PET could be a differential diagnosis was limited. Of the 12 cases where there was data on blood pressure, PET was not a differential for nine (75%) but could have been a differential for three (25%). For six (40%) cases out of the 15 with enough information, AFLP was a differential diagnosis–either as stated in the article or it met the Swansea Criteria for diagnosis [56] with six or more of the criteria met. Fourteen cases were suitable for assessing if HELLP was a differential diagnosis, and for nine (64%) it was a differential, either due to being stated by authors or according to the cutoff values we set prior to data extraction (S3 Text), informed by the Mississippi and Tennessee classifications [57]

**Fetal and maternal outcomes.**   There were four cases of maternal death out of 35, with most patients surviving an infection of leptospirosis in pregnancy. There were 20 cases of fetal demise including stillborn, IUFD (intrauterine fetal death) and miscarriages suggesting that adverse fetal outcomes are more common. There were 15 live births, eight of which were preterm and seven were term births. There was one case of a neonatal death, where the infant died 14 hours after the onset of moderate icterus, with confirmed congenital leptospirosis. The mother of this infant also died following childbirth [38].

We further analysed the fetal outcomes data to identify the maternal clinical characteristics that could be associated with fetal or neonatal death. The results are shown in Table 7. Due to small numbers we were unable to derive a meaningful result, but a higher proportion of pregnant women who presented with leptospirosis in the second trimester had adverse fetal outcomes compared with women who presented in the third trimester (p<0.1).

**Table 7.  Relationship between maternal clinical characteristics and adverse fetal outcomes.**

| Mother's clinical characteristics | Live Birth with no adverse outcomes Frequency (%) | Fetal or Neonatal Death Frequency (%) | p value* |
|---|---|---|---|
| **Liver Function Tests (LFTs)** | | | |
| Normal | 1(6.25) | 2(18.18) | 0.549 |
| Deranged | 15(93.75) | 9(81.82) | |
| **Bilirubin levels** | | | |
| Normal | 4(25.00) | 5(38.46) | 0.688 |
| Hyperbilirubinaemia | 12(75.00) | 8(61.54) | |
| **Coagulopathy** | | | |
| Absent | 3(33.33) | 0(0.00) | 0.497 |
| Present | 6(66.67) | 4(100.00) | |
| **Gestational age at presentation** | | | |
| 2nd Trimester | 4(25.00) | 11(57.89) | 0.087 |
| 3rd Trimester | 12(75.00) | 8(42.11) | |

*P-value for Fisher's Exact test for difference in proportion

### Antibiotics used for treatment of leptospirosis in pregnancy

Only one observational study had data on treatment options for leptospirosis in pregnancy, and this was McGready et al's study which suggested using Azithromycin for ambulant and febrile patients with leptospirosis in pregnancy, and adding in Ceftriaxone and Metronidazole if severely ill [18]. This study was located on Thai-Burmese border and can only be taken as guidance if in keeping with local sensitivities, so all suggestions for antibiotic therapy will need to take into consideration local antimicrobial guidance.

Table 8 summarises antibiotic use from the case reports and series on leptospirosis in pregnancy. From the case reports, there were a range of antibiotics used, with antibiotics given in 25 of the 35 cases. There were only 10 cases where it was not derivable if antibiotics were used, so this could include cases where antibiotics might have been used but not reported. Eighteen of the cases allowed enough time to see if the clinical picture improved or deteriorated and there were no reported adverse effects from the antibiotics of choice. Penicillins were the most commonly used antibiotic. For two patients, when leptospirosis was suspected or diagnosed postnatally, they were started on Doxycycline. Other antibiotics that were used included Co-amoxiclav, Meropenem and Dihydrostreptomycin. From the available literature, it was not possible to ascertain the effect of antibiotics on disease prognosis or maternal and fetal outcomes.

## Discussion

We extracted data from eight observational studies, four of 'poor' quality, three of 'fair' quality, and one of 'good' quality, highlighting the distinct lack of good quality observational studies exploring leptospirosis in pregnancy. We also identified 35 reported cases of leptospirosis in pregnancy from 21 case reports and three case series, of which there were four 'poor' quality case reports, nine 'fair' quality case reports, nine of 'good' quality and two 'fair' quality case series. We calculated the incidence of leptospirosis in pregnancy to be 1.3 per 10,000 in women presenting with fever or with jaundice, which falls in the lower end of the range of overall incidence of leptospirosis in the general population, estimated to be 0.01 to 97.5 annual cases per 10,000 population in another systematic review [3]. Incidence in the pregnant population could be higher in endemic areas. The seroprevalence of leptospirosis in healthy pregnant women across 10 Caribbean countries was 18.6% ± 3.6, based on the presence of IgG antibodies, which can be a useful guide of seroprevalence of leptospirosis in pregnant populations in other endemic areas for which we do not have enough data. There is evidence to suggest that leptospirosis can occur in both rural and urban areas. All the identified cases

**Table 8. Antibiotic therapy.**

|  |  | Frequency n = 35* | %* |
|---|---|---|---|
| **Antibiotics** | Ampicillin | 11 | 44% |
|  | Amoxicillin | 1 | 4% |
|  | Penicillin–unspecified | 3 | 12% |
|  | Penicillin G | 3 | 12% |
|  | Azithromycin | 1 | 4% |
|  | Ceftriaxone | 3 | 12% |
|  | Other antibiotic therapy | 3 | 12% |
|  | NA | 10 |  |

*Percentages were calculated after excluding missing data (NA); NA = Not Available

presented with symptoms in the second or third trimester, but adverse fetal outcomes were more common in patients who presented in the second trimester. Leptospirosis in pregnancy can present as a spectrum of disease, including the following signs and symptoms: fever, headache, generalised myalgia, and jaundice. Additionally, it can lead to congenital infection, miscarriage, IUFD (intrauterine fetal death), stillbirth as well as neonatal and maternal death. Biochemically, leptospirosis affects pregnancy much like it does the non-pregnant population, and can cause a leucocytosis, deranged liver function tests and coagulopathies. The aforementioned presentation in pregnancy, due to similarities with pregnancy related disorders, can mask a diagnosis of leptospirosis and prevent it from being in the immediate shortlist of differential diagnoses to health professionals. However whilst a diagnosis of leptospirosis is formally confirmed with MAT and ELISA tests, often there can be an indicator if a febrile patient in an endemic area has been exposed to leptospirosis from the initial history. Finally, regarding treatment, despite tetracyclines being generally avoided in pregnancy, there is a long list of antibiotics that are suitable for use in pregnancy that can be effective against leptospirosis and should be used if suspected in accordance with local sensitivities.

### 'Ask early'—Broaden differential diagnoses

Given that in most cases there was a clue or indication about exposure that could be gained from the history, this could be a low-cost-high-yield recommendation for clinicians looking after pregnant patients who present with any of the following symptoms: fever/abdominal pain/headache/myalgia/jaundice, or the aforementioned biochemical disturbances, in either endemic areas or in areas where there have been concerns regarding water and sanitation. This may be either longstanding, or emerging due to recent disasters or floods for example. In clinical practice, there is already a cluster of questions that clinicians are taught to ask when suspecting infectious disease, ranging from sexual history to travel history. Questions screening for leptospirosis exposure about raw food ingestion, exposure to farming environments, poor hygiene conditions could be added to this existing cluster of questions in a history taken by a clinician. It is particularly important for clinicians who primarily deal with pregnancy-associated pathology to broaden their differential diagnoses, especially when leptospirosis can firstly be subtle, and secondly treatable. One of the case reports stated that, 'As the main focus was on pregnancy and the probable diagnosis was HELLP syndrome, the other differential diagnoses were not adequately investigated' [39]. This is the testament to the importance of keeping differential diagnoses as broad as possible, particularly in a specialised profession. Furthermore, if not asked at the initial encounter, and the pregnancy is later ended (irrespective of perinatal outcome), and patient continues to deteriorate despite no longer being pregnant, this should raise suspicion that the main underlying pathology is not a pregnancy-associated condition, and should trigger consideration of broader differentials, including leptospirosis in an endemic area.

### 'Treat early'—Increase index of suspicion and combine with rapid field diagnosis and early treatment

The first recommendation to 'ask early' and screen for leptospirosis exposure as part of basic history taking when screening for infectious diseases, must be combined with the recommendation to 'treat early.' Whilst Doxycycline is commonly avoided in pregnancy due to the presumed tetracycline effect on skeletal development and permanent teeth discolouration [58], penicillins are known to be effective against leptospirosis [59] and also safe in pregnancy. If leptospirosis is within the differential, a broad-spectrum penicillin in keeping with local sensitivities could provide adequate coverage for leptospirosis and other infectious diseases on the

differential list. There is no clear antibiotic identified as being the most effective for leptospirosis treatment [59]. It is thus a recommendation to firstly raise the index of clinical suspicion for leptospirosis in relevant areas, and secondly to treat early with antibiotics according to local sensitivities and microbiology guidance. We appreciate that the benefits of treating a patient with suspected but not confirmed leptospirosis with antibiotics must be weighed up with the risks of antimicrobial resistance—this must be a risk assessment that each clinician considers. In endemic areas, a clinical algorithm for management and treatment of a febrile pregnant patient that is location-specific may be a useful aid to help clinicians with the decision-making process. A clinical algorithm also combines the opportunity to incorporate management of other local causes of febrile illness. Furthermore, this highlights the need for a rapid field-based test. The combination of our first recommendation of 'asking early' and assessing for likelihood of leptospirosis exposure, with a rapid field-based test that confirms diagnosis prior to commencing antibiotics, would reduce the risk of antibiotic resistance without compromising patient safety through delayed or missed opportunities to diagnose and treat.

### 'Report well'—Quantity and Quality of Evidence and Data

In order to change the landscape of evidence that exists surrounding leptospirosis in pregnancy, there is a need for more 'good' quality data from larger epidemiological studies. Studies on leptospirosis need to be inclusive of pregnant patients. If a clinician is writing a case report, this needs to be reported well—for example include numerical data of investigations, report negative symptoms the patient didn't have, include the weight of the infant if relevant and share trends such as how long was the patient febrile for and the trends of deranged biochemical markers.

### Strengths and Limitations

In keeping with the above recommendation, the most overwhelming limitation of this review is the limited amount of data and the quality of the studies available. Whilst leptospirosis is formally classified as an NTD, and we expect the associated challenges with this, this does not explain why pregnant people are often excluded from relevant studies, or the quality of many case reports being 'poor.' Nevertheless, this is the first systematic review on leptospirosis in pregnancy undertaken using a thorough methodology with a comprehensive search of published and unpublished literature. Synthesising data primarily from case reports is associated with significant bias, but in the context of an NTD and limited data, it can provide us with key information in diagnosing, treating and reducing mortality and morbidity due to leptospirosis in pregnancy.

### Conclusion

We know that the combination of the climate changing and leptospirosis being an emerging infectious disease [50,51] means that if as predicted, we see more outbreaks, we need a sound understanding of how leptospirosis affects pregnant populations and how it can be treated. This systematic review identified and reviewed all available literature on this topic to make the following three key recommendations: ask early, treat early, and report well. This refers to the incorporation of screening questions regarding possible leptospirosis exposure in the process of expanding differentials beyond pregnancy-related conditions, combining a low threshold for suspecting leptospirosis in endemic areas with early diagnosis and appropriate antibiotic therapy and finally, the call for more high quality evidence on this topic.

## Supporting information

**S1 Text. PROSPERO Registration.** PROSPERO: CRD42020151501. https://www.crd.york.ac.uk/prospero/display_record.php?RecordID=151501.
(PDF)

**S2 Text. Search Strategy.**
(DOCX)

**S3 Text. Coding for Data Extraction from Case Reports and Series.**
(DOCX)

**S1 Table. Quality assessment tool for case reports and case series.**
(DOCX)

**S2 Table. Quality assessment tool for cohort and cross-sectional studies.**
(DOCX)

**S3 Table. Summary of ongoing studies.**
(DOCX)

**S4 Table. Record of all included Case Reports and Series.**
(DOCX)

**S5 Table. Record of excluded articles.**
(DOCX)

## Acknowledgments

We acknowledge the help of Sinaida Cherubin, Gracia Fellmeth, Ronan Lyne, Daniel Robles Ortiz and Nicola Grasso in helping with the translation of articles. We acknowledge the assistance of Joseph Hornby in creating Figs 1 and 2.

## Author Contributions

**Conceptualization:** Sujitha Selvarajah, Manisha Nair.

**Data curation:** Sujitha Selvarajah, Shaolu Ran.

**Formal analysis:** Sujitha Selvarajah, Manisha Nair.

**Funding acquisition:** Manisha Nair.

**Investigation:** Sujitha Selvarajah, Shaolu Ran.

**Methodology:** Sujitha Selvarajah, Shaolu Ran, Nia Wyn Roberts, Manisha Nair.

**Project administration:** Sujitha Selvarajah.

**Resources:** Sujitha Selvarajah, Manisha Nair.

**Software:** Sujitha Selvarajah.

**Supervision:** Manisha Nair.

**Visualization:** Sujitha Selvarajah.

**Writing – original draft:** Sujitha Selvarajah.

**Writing – review & editing:** Sujitha Selvarajah, Shaolu Ran, Nia Wyn Roberts, Manisha Nair.

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
