## [Decision Letter · Decision Letter 0]

6 Jul 2021

Dear Dr Selvarajah,

Thank you very much for submitting your manuscript "Leptospirosis in Pregnancy: A Systematic Review" for consideration at PLOS Neglected Tropical Diseases. As with all papers reviewed by the journal, your manuscript was reviewed by members of the editorial board and by several independent reviewers. The reviewers appreciated the attention to an important topic. Based on the reviews, we are likely to accept this manuscript for publication, providing that you modify the manuscript according to the review recommendations. 

Sincerely,

Arnau Casanovas-Massana, PhD

Associate Editor

Justin Remais

Deputy Editor

Reviewer's Responses to Questions

**Key Review Criteria Required for Acceptance?**

**Methods**

-Are the objectives of the study clearly articulated with a clear testable hypothesis stated?

-Is the study design appropriate to address the stated objectives?

-Is the population clearly described and appropriate for the hypothesis being tested?

-Is the sample size sufficient to ensure adequate power to address the hypothesis being tested?

-Were correct statistical analysis used to support conclusions?

-Are there concerns about ethical or regulatory requirements being met?

Reviewer #1: -Are the objectives of the study clearly articulated with a clear testable hypothesis stated? YES

-Is the study design appropriate to address the stated objectives? YES

-Is the population clearly described and appropriate for the hypothesis being tested? YES

-Is the sample size sufficient to ensure adequate power to address the hypothesis being tested? YES

-Were correct statistical analysis used to support conclusions? YES

-Are there concerns about ethical or regulatory requirements being met? NO

Reviewer #2: -Are the objectives of the study clearly articulated with a clear testable hypothesis stated? Yes.

-Is the study design appropriate to address the stated objectives? Yes.

-Is the population clearly described and appropriate for the hypothesis being tested? Yes.

-Is the sample size sufficient to ensure adequate power to address the hypothesis being tested? Yes.

-Were correct statistical analysis used to support conclusions? Yes.

-Are there concerns about ethical or regulatory requirements being met? Yes.

**Results**

-Does the analysis presented match the analysis plan?

-Are the results clearly and completely presented?

-Are the figures (Tables, Images) of sufficient quality for clarity?

Reviewer #1: -Does the analysis presented match the analysis plan? YES

-Are the results clearly and completely presented? YES

-Are the figures (Tables, Images) of sufficient quality for clarity? YES

Reviewer #2: -Does the analysis presented match the analysis plan? Yes.

-Are the results clearly and completely presented? Yes.

-Are the figures (Tables, Images) of sufficient quality for clarity? Yes.

**Conclusions**

-Are the conclusions supported by the data presented?

-Are the limitations of analysis clearly described?

-Do the authors discuss how these data can be helpful to advance our understanding of the topic under study?

-Is public health relevance addressed?

Reviewer #1: -Are the conclusions supported by the data presented? YES

-Are the limitations of analysis clearly described? More emphasis on the lack of a field based diagnostic tool could be provided. The authors have been clear in stressing that pregnant women MUST be included in fever studies.

-Do the authors discuss how these data can be helpful to advance our understanding of the topic under study? Yes and a bit more could be added here on fever algorithms for diagnosis and treatment in pregnancy. These have to made locally due to different epidemiology and different access to diagnostic tools.

-Is public health relevance addressed? Yes

Reviewer #2: -Are the conclusions supported by the data presented? Yes.

-Are the limitations of analysis clearly described? Yes.

-Do the authors discuss how these data can be helpful to advance our understanding of the topic under study? Yes.

-Is public health relevance addressed? Yes.

**Editorial and Data Presentation Modifications?**

Reviewer #1: Thank you for the invitation to review this ‘first’ systematic review of leptospirosis in pregnancy. The authors have made the best of limited data and produced a helpful review. I have only minor comments overall.

The ‘treat early’ recommendation needs to be tempered by risk benefit in relation to antimicrobial resistance. A low tolerance to treat in pregnancy results in overtreatment varying with the epidemiology. This is also supported by the findings in the section of global burden in pregnant women with symptoms: Pooling together two datasets, (Studies 1,3; Table 1 [20,22]) we calculated the incidence of leptospirosis in febrile pregnant patients to be 1.3 per 10,000 (95% CI 0.5 to 2.9 per 10,000). The calculated incidence of leptospirosis in pregnant patients with jaundice, pooling two datasets (Studies 2, 6; Table 1 - [21, 24]; Table 1) was comparable, 1.3 per 10,000 (95% CI 0.2 to 4.6 per 10,000). With such a low incidence in 'symptomatic' women overtreatment (without supportive diagnostics) is inevitable and not without consequence.

It would be helpful for the reader to have some context to the estimation of incidence. For PW the estimate is 1.3 per 10,000 in women presenting with fever or jaundice, which compared to an overall incidence estimate of 0.01 to 97.5 annual cases per 10,000 population [ref PLoS Negl Trop Dis. 2015 Sep; 9(9): e0003898. Published online 2015 Sep 17. oi: 10.1371/journal.pntd.0003898 Global Morbidity and Mortality of Leptospirosis: A Systematic Review Federico Costa,# 1 , 2 , 3 José E. Hagan,# 1 , 3 Juan Calcagno, 1 Michael Kane, 4 Paul Torgerson, 5 Martha S. Martinez-Silveira, 1 Claudia Stein, 6 Bernadette Abela-Ridder, 7 and Albert I. Ko 1 , 3 ,*

In Q 12 of the quality assessment tool “Were the outcome assessors blinded to the exposure status of participants?” How was this interpreted: ‘outcome’ of the acute febrile episode or ‘outcome’ of the pregnancy? McGready et al were blind to the outcome of the fever diagnosis as leptospirosis by MAT and Scrub typhus serology (and PCR) were done well after study completion (they were not available in the study area); while malaria was instant, dengue same day, and urine and blood culture results were known within a few days. Apart from fever and miscarriage or delivery on the same day as enrolment to the study (very rare), most cohort studies with an acute febrile episode and enrolment in pregnancy are blind to the birth outcome.

Doxycycline: consider rephrasing from ‘cannot be used in pregnancy’ in the discussion, to ‘remains an option in pregnancy’, or other less definitive phrase – see EXPERT OPINION ON DRUG SAFETY, 2016, VOL. 15, NO. 3, 367–382 http://dx.doi.org/10.1517/14740338.2016.1133584

We should not discourage asking for risk factors in any febrile episode in pregnancy. There was no mention of conjunctival suffusion frequently a clue to help differentiate leptospirosis from malaria, dengue, scrub typhus and other rickettsial illness, ehrlichoiosis and acute viral infections including influence (and now COVID-19). Maybe this could be mentioned as a deficiency of the studies? How many mention or report it? Clinical acumen was very low in McGready et al [after excluding malaria cases]: treat early would have resulted in 4 in 5 suspected women receiving antibiotics who don’t need them [clinical accuracy was 16.7% (1/5)].

Add the units of the seroprevalence: seroprevalence of 18.6 +/- 3.6 overall? Assume it is %.

Consider suggesting a fever algorithm for diagnosis and treatment of pregnant women: Each place in the world that treats pregnant women with fever requires an algorithm (based on what they have available to them) to support optimal care [perhaps there are sufficient cohort studies that could collectively develop this – they would vary slightly based on local epidemiology]. It is not foolproof – in McGready et al there was a death in a treated patient and no cause was found by any of the diagnostics available. A fever algorithm embodies your other suggestions for: ‘ask’, ’good clinical exam’, ‘use lab diagnostics’…treat early. A diagnosis by exclusion can also be helpful to minimize overuse of antibiotics. 

Azithromycin in McGready et al was used in the pregnancy fever algorithm because at the time it was thought a safe option for treatment of scrub typhus in the area but also covered leptospirosis which is less prevalent. 

The manuscript discussion could stress/highlight the inadequate laboratory diagnostics for confirmation of this widespread zoonosis and the very great need to for a field based tool. RDT for dengue and malaria are very useful in practice (both these diseases don't need antibiotics). 

Very minor

Reference 3 – it says website? 

Missing punctuation: communicable diseases[1] needs a full stop

Check your formatting of references: sometimes before punctuation and other times after e.g after “over humanity. [2]” and before:” [6,10].”

Reviewer #2: Dear Editor,

This manuscript “Leptospirosis in Pregnancy: A systematic review by Selvarajah et al. is an excellent, well-written article. Given the complexity involved, the author has produced a number of positive and welcome outcomes, which offers a useful overview of current research and policy and the resulting bibliography which provides a very useful resource for current practitioners. It is one to which the author(s) have made significant contributions.

Please note that I am not familiar with the details of the Meta analysis so cannot comment on their appropriateness and accuracy.

I have no hesitation in recommending that it be accepted for publication after a few typos and other minor details have been attended to.

Best regards,

Vanaporn Wuthiekanun

**Summary and General Comments**

Reviewer #1: not applicable

Reviewer #2: Comments to the authors

This manuscript “Leptospirosis in Pregnancy: A systematic review by Selvarajah et al. is an excellent, well-written article. Given the limitation involved, the author has produced a useful overview that provides a very useful resource for current practitioners. The documentation and discussion appear descriptively appropriate. After careful consideration, the manuscript will likely to be suitable for publication if it is revised to address the minor points below.

Minor comments

Table 1, Study 3 [22], Quality: I think it should be changed “Good-9” to “moderate-9” according to the first sentence of page 18 (Line 183-185, The majority of observational studies had a moderate to high level of bias, being of ‘fair’ to ‘poor’ quality, except for one cross-sectional study (Study 1, Table 1 and 2) which was graded to be of ‘good’ quality.).

Edit MS for space and double full-stop (such as at line 84, 205, 434 and etc).

Check reference no. 3, line 491.

Typo

1. Abstract, introduction, 3rd sentence “was” should be “were”

2. Line 43; “underreported” to “underreporting”

3. Line 51; “to efficacy” to “ to the efficacy”

4. Line 96; “the sparsity of “ to “sparse”

5. Line 218; “Carribean” to “Caribbean”

6. Line 272; it is better to use the full name for the first time such as Microscopic Agglutination Test (MAT).

7. Line 417: “state” to “states”

PLOS authors have the option to publish the peer review history of their article (what does this mean?). If published, this will include your full peer review and any attached files.

Reviewer #1: No

Reviewer #2: No

Figure Files:

Data Requirements:

Reproducibility:

References

---

## [Editor Report · Decision Letter 1]

20 Aug 2021

Dear Dr Selvarajah,

We are pleased to inform you that your manuscript 'Leptospirosis in Pregnancy: A Systematic Review' has been provisionally accepted for publication in PLOS Neglected Tropical Diseases.

Best regards,

Arnau Casanovas-Massana, PhD

Associate Editor

Justin Remais

Deputy Editor

---

## [Editor Report · Acceptance letter]

9 Sep 2021

Dear Dr Selvarajah,

We are delighted to inform you that your manuscript, "Leptospirosis in Pregnancy: A Systematic Review," has been formally accepted for publication in PLOS Neglected Tropical Diseases.

Best regards,

Shaden Kamhawi

co-Editor-in-Chief

Paul Brindley

co-Editor-in-Chief
